# Earthquake Building Damage Detection based on Synthetic Aperture Radar Imagery and Machine Learning

Anirudh Rao[1], Jungkyo Jung[2], Vitor Silva[1], Giuseppe Molinario[3], and Sang-Ho Yun[4, 5, 6]

[1]Global Earthquake Model Foundation, Pavia, Italy
[2]Jet Propulsion Laboratory, California Institute of Technology, CA, USA
[3]World Bank Group, Washington DC, USA
[4]Earth Observatory of Singapore, Nanyang Technological University, Singapore
[5]Asian School of the Environment, Nanyang Technological University, Singapore
[6]School of Electrical and Electronic Engineering, Nanyang Technological University, Singapore

**Correspondence:** Anirudh Rao (anirudh.rao@globalquakemodel.org)

**Abstract.** This article presents a framework for semi-automated building damage assessment due to earthquakes from remote sensing data and other supplementary datasets, while also leveraging recent advances in machine-learning algorithms. The framework integrates high-resolution building inventory data with earthquake ground shaking intensity maps and surface-level changes detected by comparing pre- and post-event InSAR (interferometric synthetic aperture radar) images. We demonstrate the use of ensemble models in a machine-learning approach to classify the damage state of buildings in the area affected by an earthquake. Both multi-class and binary damage classification are attempted for four recent earthquakes and we compare the predicted damage labels with ground truth damage grade labels reported in field surveys. For three out of the four earthquakes studied, the model is able to identify over fifty percent or nearly half of the damaged buildings successfully when using binary classification. Multi-class damage grade classification using InSAR data has rarely been attempted previously, and the case studies presented in this report represent one of the first such attempts using InSAR data.

## 1 Introduction

In the immediate aftermath of an earthquake, timely and reliable assessment of the impact in terms of the damage sustained by built assets and the associated repair and replacement costs assumes a crucial role in the strategic organization and prioritization of the response and recovery efforts. Selection of a particular approach for damage and loss evaluation typically involves a trade-off between the level of details collected and the timeliness of evaluation.

Within the public sector, estimation of damage and loss after a natural disaster is typically performed through field missions using paper forms, or more recently, data capture tools on tablet and mobile devices (Esri, 2021; United Nations Development Programme, 2021). This approach requires the mobilization of technical experts to the affected areas. While field-surveys still remain the most accurate solution, they can be highly resource intensive and time-consuming for disasters affecting large areas. Lack of available and experienced personnel can also be an issue especially for large scale events. Furthermore, for some disasters it might not even be possible to access some of the affected regions due to disruption of the transportation network.

Completing such detailed damage assessments for moderate to large events can take weeks, if not months (Silva et al., 2018). For instance, collection and compilation of building-level damage data after the 2015 M7.8 Gorkha earthquake and landslides in Nepal continued for over seven months (Silva et al., 2018).

Remote sensing data offers advantages over ground-inspection in terms of its collection speed and spatial coverage, particularly when the emphasis is less about producing a detailed and highly accurate building-level inventory of damage, but on providing rapid information about the potential locations, extents, and severity of damage. In recent years the improvement of the level of detail, spatial resolution, and reduced latency of Earth-observation (EO) data has encouraged a variety of applications in disaster damage assessment. Dell'Acqua and Gamba (2012); Dong and Shan (2013); Plank (2014); Ge et al.

(2020) provide thorough reviews of current methods for earthquake-induced building damage detection using EO data. Optical EO data is currently available at sub-meter spatial resolution from several satellites, making it particularly appealing for building-level damage detection methods. Optical imagery is also conducive for applications involving visual interpretation, such as crowd-sourced damage labelling. There are presently a few rapid damage mapping services in operation which use primarily optical EO data as the basis for damage assessment. These include the Operational Satellite Applications Programme

(UNOSAT) of the United Nations (United Nations Institute for Training and Research, 2003) and the Copernicus rapid damage mapping service supported by the European Commission (Copernicus Emergency Management Service and The European Commission, 2012). These damage assessments by UNITAR and Copernicus need significant manual effort to scan the raw optical imagery covering the affected area for collapsed buildings, signs of debris or other visible damage, and cracks in bridges and other infrastructure elements.

As an alternative, radar EO data is provided by active airborne or space-borne radio detection and ranging sensors. These sensors emit pulses of microwave radiation towards a target on the Earth's surface, which reflects back some of the emitted energy. Synthetic Aperture Radar (SAR) is a technology used to exploit the continuous transmission and reception of radar pulses to and from a radar imaging system mounted on a moving platform, such as an airplane or a satellite. Processing of the signals from the multiple pulses received from the same target but at different relative locations of the sensor can in

effect, help create a larger "synthetic aperture" that can allow capturing images at a much higher resolution than a similar stationary antenna. Light detection and ranging (LiDAR) sensors work in a manner similar to radar sensors, using laser pulses instead of radio pulses. For damage detection applications, LiDAR data is typically obtained through airborne sensors that can collect 3D data in the form of point clouds. Pre-event LiDAR data are typically not available for most events, making it challenging to attempt change detection. Thus, nearly all studies involving damage detection using LiDAR data involve only

the post-earthquake data.

A significant advantage offered by SAR over optical EO and LiDAR is that SAR data can be obtained even in poor-light conditions including at night, can penetrate through ground-level obstacles, and is independent of cloud cover. Heretofore, most of the SAR-based methods described in the comprehensive review undertaken by Ge et al. (2020) attempt to infer changes and predict damage at the block level, rather than at the building level. One reason for this has been the limited availability of very

high-resolution SAR data. However, meter-level spatial resolution is now offered by several SAR satellites, including ALOS-2, COSMO-SkyMed, TerraSAR-X, and TanDEM-X, which makes building level damage detection promising, especially when

ancillary datasets such as digitized building footprint layers are available for use in conjunction with high resolution SAR data (e.g., Bai et al. (2017); Ge et al. (2019); Miura et al. (2019); Natsuaki et al. (2018)).

Meanwhile, recent advances in machine learning (ML), with high-performance open-source libraries for training and evaluating models have led to an increasing body of work that aims to use learning algorithms to predict damage following earthquakes. Broadly, these efforts can be classified into the following four categories: (i) ML models using building attributes and geophysical features alone (e.g., Mangalathu et al. (2020); Roeslin et al. (2020)), (ii) ML models using optical EO data (eg. Ji et al. (2018, 2019, 2020); Lee et al. (2020); Xu et al. (2019); Tilon et al. (2020); see Nex et al. (2019) for an extensive review), (iii) ML models using SAR data alone (e.g., Wieland et al. (2016); Bai et al. (2017); Stephenson et al. (2021)), and (iv) ML models using SAR EO data in conjunction with building attributes and geophysical features (e.g., Moya et al. (2018a, b); Xie et al. (2020)).

Roeslin et al. (2020) evaluated the performance of various ML classification algorithms to classify building damage for the 2017 Puebla, Mexico earthquake, based on input features including structural attributes of the buildings and seismic demand in terms of maximum spectral acceleration. The random forest model was found to have the highest relative accuracy amongst the models evaluated, being able to correctly identify 78% of the damaged buildings in the test set. Mangalathu et al. (2020) also evaluated the performance of different ML classification algorithms to classify building damage from the 2014 South Napa earthquake, and indicated that the random forest algorithm provides the best performance amongst the evaluated techniques. The input features included distance from the fault, spectral acceleration, and structural attributes of the buildings. However, even the random forest algorithm was correctly able to identify only 12.5% of the red-tagged buildings, though it was able to correctly classify 79% of the yellow-tagged buildings. Nex et al. (2019) provided a comprehensive summary of the state-of-the-art on earthquake building damage detection using deep learning methods, mostly based on convolutional neural networks (CNNs), with optical imagery obtained by remote sensing satellite or airborne sensors.

The use of machine learning algorithms in conjunction with SAR data for earthquake damage detection is a recent development, and relatively fewer studies are found in the literature compared to optical EO data-based studies that employ machine learning. Wieland et al. (2016) evaluated the application of a Support Vector Machine (SVM) classifier to identify changes in single- and multi-temporal X- and L-band SAR images from TerraSAR-X and ALOS PALSAR, and used their classifier to detect damage from the 2011 Tohoku earthquake and tsunami. While the single-image approach that uses only the post-image SAR data yielded reasonable results, the multi-temporal approach demonstrated a greater performance. The authors report that the SVM classifier performs well for binary classification, but performance degrades when trying to classify damage into multiple grades. Bai et al. (2017) in their research tried to shed some light on the difference in the building damage mapping performance when using multi-temporal or only post-event SAR images in the framework of machine learning. The K-Nearest Neighbours learning algorithm was selected as the preferred classifier, as it showed the best performance in evaluation. Using the 2016 Kumamoto earthquake as a case study, the authors indicate a prediction accuracy of 40.1% for damaged buildings when only post-event SAR images were used, and a 38.9% accuracy for identifying damaged buildings when multi-temporal SAR images were used.

Moya et al. (2018b, a) proposed a new approach for the classification of collapsed buildings in the aftermath of a disaster based on SAR imagery, the spatial distribution of hazard, and a set of fragility functions. The method was applied to the 2011 Tohoku earthquake and tsunami for collapsed building detection using two TerraSAR-X images (one pre-event image, and one post-event image) of the coastal area of Miyagi Prefecture. Their binary classification into collapsed and non-collapsed buildings was compared with field survey damage data. Depending on the fragility function used, the method was able to correctly classify between 80.4–92.7% of the buildings that were completely washed away by the tsunami, whereas 61.2–69.8% of the buildings categorized as collapsed were actually washed away.

In this study, we build upon the aforementioned work and propose a supervised ML framework for building-level earthquake damage assessment using high-resolution SAR data, which we combine with high-resolution building inventory datasets and earthquake ground shaking intensity maps to classify buildings into different damage states. We attempt both binary damage classification, as well as multi-class damage classification for four recent earthquakes using ensemble models in a machine-learning approach. Multi-class damage grade classification using SAR EO data has rarely been attempted, as evidenced by the previously presented literature. Thus, the case studies presented in this report represent one of the first such attempts using SAR data. Comparing the predicted damage states with ground truth data obtained through field surveys allows us to assess the accuracy of the ML model for the different case study earthquakes.

## 2 Data, Methodology, and Study Areas

The key elements of the framework include the SAR-derived damage proxy map (DPM) for the event, a ground shaking intensity map such as the ShakeMap published by the U.S. Geological Survey (USGS), and a building inventory layer which includes at minimum the building footprints in the affected area. Where available, additional building attributes can also be incorporated into the framework. Finally, ground truth damage grade labels for the buildings in the affected area are needed to train and test the machine learning model for damage classification. The proposed approach is designed to be compatible with both detailed exposure information (building-by-building data) or proxy exposure information mapped on to a building footprint layer (for regions where building-specific data are not available), although its prediction accuracy is enhanced when building-specific data are used. Whereas the illustrative application case studies in this paper focus on implementing and testing this framework for earthquake related damage, the proposed framework adopts a modular approach, so that it can be extended to other natural hazards by tailoring the hazard-specific parts whilst using the core building blocks that would be common to all hazards. The various input datasets used by the framework and the processing steps are described in this section.

### 2.1 Input data

### 2.1.1 SAR-derived Damage Proxy Maps

The Advanced Rapid Imaging and Analysis (ARIA) project for natural hazards, a joint effort of NASA's Jet Propulsion Laboratory (JPL) and California Institute of Technology (Caltech), has developed a product called "Damage Proxy Maps" (DPMs),

based on the methodology described by Yun et al. (2015a) for comparing InSAR coherence maps prior and subsequent to a damaging event. The product is termed as a proxy map because it is derived from EO data and based on limited ground truth if any. The ARIA team makes the DPMs available in standard GeoTIFF raster format, where each pixel typically measures about 30 meters across. Raw DPM pixel values range from 0 to 1, where higher values are indicative of increasingly larger surface change.

Automation of the DPM generation using SAR data from Sentinel-1, ALOS-2, and COSMO-SkyMed missions (i.e., two C-band, one L-band, four X-band satellites, respectively) helps achieve a significant reduction in satellite overpass latency. On average, at a latitude of 36°, one of the seven SAR satellites can be expected to overpass a disaster-hit area in about 10 hours, and DPM products can be potentially generated for an earthquake event within 24 hours of data acquisition. Figure 1 shows the DPM generated by the ARIA team for the March 2020 Mw5.3 Zagreb earthquake in Croatia using multi-temporal interferometric coherence of Copernicus Sentinel-1 SAR data (Jung et al., 2016). This event is one of the four earthquakes that were considered to test the framework presented herein.

DPMs have also been previously used in earthquake damage assessment frameworks. For instance, Loos et al. (2020) used the ARIA DPM product as one of the inputs in their geospatial data integration framework to assess post-earthquake mean damage ratios. The public availability of DPMs for a large number of significant damage-causing earthquakes starting from the 2014 M6.0 South Napa earthquake, its qualitative validation in past events (Yun et al., 2015b; Sextos et al., 2018), and the possibility of generating DPMs for new events within a few days of the event, makes the DPM product a logical choice for the earth-observation proxy in the damage assessment framework proposed in this study.

### 2.1.2 Ground shaking intensity maps (ShakeMaps)

For significant earthquake events worldwide, the USGS Earthquake Hazards Program in partnership with regional seismic networks, distributes maps of shaking intensities in near real-time. This product is named ShakeMap, and it is released through the USGS earthquake hazards program web-portal (Wald et al., 2022). An application programming interface (API) is also provided by the USGS for accessing ShakeMap data directly through automated programs. Amongst several use cases, ShakeMaps provide rapid information about the areas that are likely to be affected by the ground shaking and the intensity of these effects. Amplitude values of ground shaking parameters are typically provided at a 1 km grid spacing. Figure 2 shows the USGS generated ShakeMap for the March 2020 Mw5.3 Zagreb earthquake in Croatia. ShakeMaps are used in the earthquake damage evaluation workflow to identify the geographical extent of the area likely to be affected by the earthquake, as well as one of the input features for the machine learning models to predict damage.

### 2.1.3 Building inventories and building footprint maps

A building footprint map is used to precisely identify the locations of the buildings in the areas affected by the earthquake. Each building footprint serves as the anchor for both the input features for the damage classification model, as well as the ground truth damage grade labels.

The OpenStreetMap (OSM) project is one source offering near-global coverage for building footprints. Building extracts from OSM can contain detailed information about building footprints and locations, and in some cases also data concerning the construction material, occupancy class, number of stories and built-up area. The OSM building inventory datasets, where available, can thus offer exposure information at a high level of detail and reliability. Although these datasets are not available with a sufficient degree of completeness at the global scale, recent initiatives harnessing advances in deep learning on high resolution satellite imagery are aiming to fill in the missing gaps (eg. Microsoft (2018); Sirko et al. (2021)).

The building damage assessment framework proposed herein is designed to make use of existing exposure datasets developed by local governments or third-party agencies, if those datasets are found to be of better quality than the OSM building extracts. Figure 3 shows the building inventory dataset compiled by the City of Zagreb, which includes not only the footprints of all buildings in the historical city center, but also information about the number of stories and the occupancy class of each building.

## 2.2 Data processing and supervised learning

### 2.2.1 Processing of the SAR-derived DPMs

The SAR-derived DPMs published by the ARIA project are used as the primary remote-sensing proxy to identify surface-level changes that are potentially attributable to earthquake-induced building damage. Since the DPMs involve inferences of damage based on variations observed on the ground surface between the before and after SAR images, the DPM tiles may indicate some apparent "damage" pixels where no buildings or infrastructure elements exist on the ground. This may happen, for instance, in areas where the vegetation changed, or due to the appearance or disappearance of vehicles between the two images. Landslides and rockfall can also lead to surface-level changes, and so can other phenomena such as building construction. Thus, while attempting to detect building damage, care needs to be exercised to limit the focus of the DPMs to locations where buildings are known to exist. Thus, we clip out parts of the DPM that are outside of built-up areas, based on building footprint maps and land-use maps. With a view to keeping the computations tractable and to reduce noise in the input vectors, we also limit the analysis to an affected area defined as the envelope of the Modified Mercalli Intensity (MMI) V contours from the ShakeMap, effectively clipping out the parts of the DPM located outside a zone where building damages are likely to have occurred due to the earthquake.

### 2.2.2 Multi-class damage classification

Supervised machine learning is employed in an attempt to predict the level of damage to the buildings. The problem presented is one of multi-class classification, in which each building needs to be classified into one out of a number of predefined set of damage states.

Deep learning techniques are more suited for structured data such as images, audio, and text corpuses with large sample sizes, while the datasets used in this study are tabular, with each row representing one building unit, and are small or medium sized (typically of the order of 1,000–10,000 samples). While both decision forest based approaches and deep neural networks could be used with tabular data, deep learning techniques perform better with large sample sizes, which is unfortunately not the

case for this study, given the limited availability of building-level damage datasets and the limited number of labelled damaged buildings within each dataset. Grinsztajn et al. (2022) conclude that for medium sized tabular data (of the order of 10,000 samples), tree-based models outperform deep learning methods, with much less computational cost. Similarly, Xu et al. (2021) also conclude that forests perform better than deep neural nets for tabular data with small sample sizes. Thus, deep learning techniques were ruled out as not being apt for the current application.

In a preliminary phase, we compared different algorithms that permit multi-class classification, including support vector machines, k-nearest neighbours, Naive Bayes, and Random Forest. Since the problem of damage classification typically involves highly imbalanced datasets, where the buildings in "no damage" state dominate the buildings in all other damage states, often by multiple orders of magnitude, all of the above classifier algorithms tended to overlearn the label with the higher number of training examples (i.e., "no damage"). The Random Forest algorithm, developed by Leo Breiman and Adele Cutler, and described in Breiman (2001) was eventually selected for the study as it allows for the assignment of weights to the training examples. The training examples in each damage class were then weighted in inverse proportion to the class frequencies observed in the input data, in order to better handle the class imbalance in the input damage datasets. The Histogram-Based Gradient Boosting classifier (Ke et al., 2017), was preferred in the cases where categorical features were present amongst the selected building attributes, in addition to purely numerical features. This was because the Histogram-Based Gradient Boosting classifier provides native categorical support, which helps avoid one-hot encoding to transform categorical features as numeric arrays.

The next step involves tuning of the hyper-parameters of the chosen classifier algorithms, where hyper-parameters are the model parameters that are not directly learnt during the training phase. Probst et al. (2019) provide a thorough overview of the hyper-parameters and tuning strategies for the random forest algorithm. Random forest algorithms have three main hyper-parameters, including the number of trees in the forest, the node size, and the number of features sampled when looking for the best split for a node. The number of features sampled at each split is set to the square root of the number of predictor variables, which Probst et al. (2019) indicate as a reasonable value for low-dimensional classification problems. Optimal values for the number of trees in the forest and the node size are obtained through an exhaustive grid search strategy, to pick the combination of hyper-parameter values that result in the best cross validation score. For all other hyper-parameters which have less of an impact on the model performance, we use the default values provided by the Python software package scikit-learn (Pedregosa et al., 2011).

The models are then trained with a 70% subset of the available data, and then the best-fit models are tested against the 30% hold-out subset. There is certainly a trade-off between using more samples for training the algorithm versus reserving sufficient samples for the test set. Using a higher fraction of the available data for training can result in over-fitting. Previous empirical studies, such as Gholamy et al. (2018), have demonstrated that using 70-80% of the data for training and reserving 20-30% of the data for testing yields optimal results in terms of improving the accuracy of the model while minimizing the tendency for over-fitting. The decision to choose a 70%/30% split for the training and testing set (say, over an 80%/20% split) was ultimately driven by the paucity of "collapse" labels in the damage datasets, where reserving only 20% of the dataset for testing would leave very few "collapse" labels in the test set to evaluate the accuracy of the fitted model.

For each building, the input 'feature vector' for the classification algorithms comprises the ground shaking intensity as measured in MMI, the highest value of the DPM pixels that fall within the building footprint, and any building attributes that may be available such as the construction material, number of stories, year of construction and slope of the terrain at the location of the building. The 'labels' that are used for the supervised learning comprise the damage grade assigned to the buildings by a structural engineering field survey following the event. In the absence of detailed field survey data for the damage labels, proxy damage labels are used, such as those generated through aerial damage survey missions. Figure 4 shows an illustration of the input feature vector and output label vector for a selected building in Zagreb.

Given the heavy imbalance typical in damage datasets, where the vast majority of buildings are in the "no damage" grade, the classifier algorithms tend to overlearn the label with the higher number of training examples. Without proper handling of this imbalance, the models tend to categorize most buildings in the test set into the "no damage" category. Several methods are available to better handle the class imbalance in the input datasets (e.g., Krawczyk et al. (2014); Feng et al. (2021)). The approach adopted in this study is to weight the training examples in each damage class in inverse proportion to the class frequencies observed in the input data. Thus, training examples involving buildings in higher damage grades would be weighted higher than those in the "no damage" grade. The trained model is then used to predict the damage grades for the subset of buildings that were intentionally left out of the training dataset, in order to gauge the prediction accuracy of the model.

### 2.2.3 Binary damage classification

While the preceding section looked at classification of building damage into multiple damage grades, an attempt is also made to classify the buildings into one of just two damage grades, i.e., "Damaged" or "Undamaged". Building damage datasets, even if they involve field inspections, are apt to contain labelling noise, due to subjectivity in assigning the various damage grades. Compressing a multi-level damage scale to a binary damage scale can help mitigate this subjectivity, and thus potentially improve the prediction accuracy, albeit at the expense of losing the finer gradation of damage levels.

The paucity of building damage data that can be used for training is one of the main challenges affecting machine learning models for damage prediction. Different countries use different methodologies and different damage scales to assess building damage following earthquakes. While the definition of the lower damage grades might differ considerably between different scales, collapse is often consistently defined. Thus, if the focus is restricted to identifying collapsed buildings from non-collapsed buildings, a wider set of events from the region can be used train the model, given that the training labels in this case coming from different events will be consistently defined.

Thus, another important reason to undertake a binary damage classification exercise is that it permits the aggregation of building damage datasets from different events into a larger training pool. Such aggregation of damage datasets is often not possible with the original multi-level damage scales, given that different damage scales are typically used for events in different countries. The procedure for creating a balanced model for binary classification is the same as that described in the preceding section for the multi-class classification model.

## 2.3 Study areas

Four recent earthquakes were used to evaluate the ML model and damage classification framework — the April 2015 M7.8 Gorkha earthquake in Nepal, the September 2017 M7.1 Puebla earthquake in Mexico, the January 2020 M6.4 Puerto Rico earthquake, and the March 2020 M5.3 Zagreb earthquake in Croatia. This section summarizes the four selected events and the datasets and sources of information used in the input vectors for each event.

### 2.3.1 25 April 2015, M7.8 Gorkha Earthquake, Nepal

The Mw 7.8 Gorkha earthquake occurred on 25 April 2015 in central Nepal, causing damage to over 750,000 buildings, of which nearly 500,000 were completely destroyed, leading to nearly 9,000 fatalities (National Planning Commission, 2015). While a complete building damage dataset covering all affected buildings in the country is available through the National Reconstruction Authority of Nepal, this dataset does not contain building footprints or geographical coordinates of the buildings. Thus, we used a combined dataset comprising building locations and damage grades covering 4,787 buildings from the Budhanilkantha municipality of the Kathmandu district, provided by the National Society for Earthquake Technology (NSET) of Nepal. This dataset also includes several building attributes for each building in the municipality, including the age, number of stories, construction type, primary occupancy class, presence of structural irregularities, and slope of the ground at the location of the building. The locations were joined to a building footprints layer for the Budhanilkantha municipality obtained from OSM.

### 2.3.2 19 September 2017, M7.1 Puebla Earthquake, Mexico

On 19 September, 2017, an earthquake of estimated magnitude Mw7.1 struck south of the city of Puebla in Mexico, causing building damage in the three states of Puebla, Morelos, and Greater Mexico City. The earthquake caused 369 fatalities, with 38 buildings completely collapsing (Roeslin et al., 2018). Buendía Sánchez and Angulo (2017) and Reinoso et al. (2021) have compiled a building damage dataset for the affected states of Puebla, Morelos, state of Mexico and Mexico City (CDMX), after collating data from multiple sources. In addition to the location and observed damage grade for each building, this dataset also contains attributes such as the structural type, the use of the building, the number of stories, and the year of construction for the majority of the buildings. Since the dataset does not contain building footprints, we joined this layer with the cadastral layer for Mexico City available from the National Institute of Statistics, Geography and Informatics (INEGI). Since the building attributes are available only for the damaged buildings and not for the undamaged buildings, we restrict the analysis to damaged buildings alone.

### 2.3.3 7 January 2020, M6.4 Puerto Rico earthquake

An earthquake of magnitude Mw 6.4 struck the southwestern region of Puerto Rico on 07 January 2020. This earthquake was the largest of a series of seismic events that started in late December of 2019, which continued into the latter part of 2020. While Puerto Rico lies in an active seismic region, the impact of this Mw 6.4 earthquake was the largest witnessed by the island

since the 1918 Mw 7.1 San Fermín earthquake. Around 335 buildings were damaged in the earthquake, of which 77 were fully destroyed (Federal Emergency Management Agency, 2020).

The building inventory for Puerto Rico was derived from FEMA's Hazus program. Input datasets involved in the creation of this inventory included building footprints from OpenStreetMap, building height data from a LiDAR survey covering the island, and the 2010 decennial census of Puerto Rico. The preliminary building damage assessment carried out by FEMA following the earthquake was used as the primary damage database for training and testing the ML model for this event. Structures were classified into three damage grades in this dataset: "Minor Damage", "Major Damage", and "Destroyed".

### 2.3.4   22 March 2020, M5.3 Zagreb earthquake, Croatia

On March 22, 2020, Zagreb was hit by an earthquake, the strongest the city has witnessed since 1880, resulting in considerable damage to public buildings and services in Zagreb and the surrounding areas. The earthquake resulted in one fatality, 26 injuries, and hundreds of people were displaced while the country was in a lockdown due to the ongoing pandemic (Government of Croatia, 2020).

The 3D model of the City of Zagreb shown in Figure 3 serves as the exposure layer in the analysis. Following the earthquake, the Croatian Interior Ministry's Civil Protection Directorate and the Croatian Chamber of Civil Engineer put out a call for the mobilization of civil engineers to help with the building damage assessment. By August 2020, over 25,000 buildings had been inspected for signs of structural and non-structural damage. A 3D view of the Lower and Upper Towns of the City of Zagreb based on the damage classifications is presented in Figure 5.

The sources for the building footprints, ground shaking intensity maps, and ground truth damage grade labels, and the availability of other building attributes for each of the four earthquakes are summarized in Table 1. The table also includes the damage grade labels that are combined into a single "Damaged" class for the binary damage classification case for each of the four earthquakes. The remaining labels are merged into a single "Undamaged" class.

## 3   Results and Discussion

We evaluate the best-fit model on both the training and test subsets using a few different performance metrics, including the precision and recall scores, the F1 score, and the balanced accuracy score (Brodersen et al., 2010). These metrics are described in brief below:

- **Precision**: Defined as the ratio $(true positives)/(true positives + false positives)$. The precision is intuitively the ability of the classifier not to label as positive a sample that is negative.

- **Recall**: Defined as the ratio $(true positives)/(true positives + false negatives)$. The recall is intuitively the ability of the classifier to find all the positive samples.

- **F1 score**: Defined as $2*(precision*recall)/(precision+recall)$. The F1 score can be interpreted as a weighted average of the precision and recall, where an F1 score reaches its best value at 1 and worst score at 0. The precision and recall scores contribute equally to the F1 score.

- **Balanced accuracy score**: This is a score intended to measure the prediction accuracy while avoiding inflated performance estimates on imbalanced datasets. It is the macro-average of recall scores per class or, equivalently, raw accuracy where each sample is weighted according to the inverse prevalence of its true class.

The results for the four case study earthquakes, for both multi-class classification and binary classification of damage are summarized below in Table 2 through Table 9. When we attempt a multi-class damage grade classification, the trained model exhibits a balanced accuracy score ranging from 0.23 for the 2017 Puebla earthquake, to 0.36 for the 2015 Gorkha earthquake and 0.40 for both the 2020 Puerto Rico and Zagreb earthquakes. The balanced accuracy scores improve significantly to 0.65 for the 2017 Puebla and 2020 Zagreb earthquakes, 0.72 for the 2020 Puerto Rico earthquake, and 0.82 for the 2015 Gorkha earthquake when we switch to a binary damage grade classification, i.e., attempting only to separate the damaged buildings from the undamaged buildings.

Figure 6 and Figure 7 show the confusion matrices for the four earthquakes for multi-class damage grade classification and binary damage classification, respectively. Table 10 provides an overall summary of the balanced accuracy scores for all of the cases considered. The recall score for the "Damaged" label in the binary damage classification scenario ranges from 0.38 for the 2017 Puebla earthquake and 0.48 for the 2020 Zagreb earthquake, to 0.58 for the 2020 Puerto Rico earthquake and 0.73 for the 2015 Gorkha earthquake, i.e., for three out of the four earthquakes studied, the model is able to identify over half or nearly half of the damaged buildings successfully when using binary classification.

We observe that for the 2015 Gorkha earthquake, for which multiple building attributes are available for both damaged and undamaged buildings, the prediction accuracies for both binary and multi-class classification are significantly higher when compared to the earthquakes where fewer or no building attributes are available for use as input features. In order to understand the impact of including the building attributes on the performance of the classifier, we also trained the ML model for this earthquake *without* using any of the building attributes and limiting the input feature vector to the MMI and DPM values alone. The precision and recall for all damage grades are lower for this reduced model compared to the results reported for the full model in Table 2 and Table 3, for both multi-class classification and binary classification respectively. The recall score for the "Destruction" damage grade drops from 0.47 for the full model to 0.31 for the reduced model in the multi-class classification task, and from 0.73 to 0.45 in the binary classification task. Similarly, the balanced accuracy score drops from 0.36 to 0.20 in the multi-class classification task, and from 0.82 to 0.59 in the binary classification task. These results clearly demonstrate the importance of including the additional building attributes in the analysis. A partial dependence analysis of the damage grade on the non-location building attribute variables for this event indicates that the building age has an impact on the damage grade, with older buildings being related to higher damage (see the Jupyter notebook for the analysis).

While four key building attributes were also available for the damaged buildings for the 2017 Puebla earthquake, the non-availability of the same for the undamaged buildings meant that a complete dataset with building attributes could not be

used for the training of the ML model. Of the four events considered in this study, the 2017 Puebla event had the smallest building damage dataset available for training the ML model. Only 219 buildings in Mexico city had a complete set of building attributes and damage labels and were also covered by the DPM and ShakeMap layers, as compared to thousands or hundreds of thousands of buildings for the other three events. While the Random Forest classification model performs well for this event in the training phase, the trained model fails to correctly identify even a single partially collapsed or totally collapsed building in the test set. Further attempts at reducing the potential overfitting of the model to the limited training data subset by adjustments to the model hyperparameters did not lead to any noticeable improvements in prediction accuracy for the event.

From Figure 6, we also observe that the true-positive prediction rates for the intermediate damage grades are lower than those for the no-damage and highest damage grades for the 2015 Gorkha earthquake and the 2020 Puerto Rico and Zagreb earthquakes. We believe that this partly stems from the fact that the existing damage scales that are widely used for field surveys of building damage, such as EMS-98 (European Sesimological Commission, 1998) do not map directly to information available through earth observation data, particularly for the lower damage grades. For instance, the first three damage grades for reinforced concrete structures according to EMS-98 involve increasing levels of cracking in the beams and columns or partition and infill walls, and buckling of the reinforcement rods. Unless this kind of damage results in debris caused by excessive spalling of the concrete cover or partial collapse of infill walls that is visible outside the structure, these damage levels as defined in EMS-98 may be challenging to identify from EO data alone. Dell'Acqua and Gamba (2012) and Cotrufo et al. (2018) both propose a building damage assessment scale tailored for optical satellite imagery and aerial imagery. However, a similar damage scale tailored for InSAR based building damage assessment is still lacking, and merits further research.

Even for the higher damage grades, a potential limitation of SAR-derived damage assessment at the building-level is that it is certainly possible to have significant seismic building damage without observing a significant corresponding change in the ground surface level. While a fully collapsed building is expected to cause higher coherence loss compared to partial collapse, field surveys of damage are likely to mark all such buildings as completely damaged or destroyed. Damage to internal walls or columns that may have severely compromised the structural integrity of the building without causing externally visible damage or collapse may not be detectable through remote sensing, as these will be out of the line-of-sight of the satellites. Storey drifts of 2% for braced steel structures or concrete shear wall structures may be technically classified as "collapsed" (eg. FEMA and ASCE (2000)) even though the structure has not physically collapsed at all, but such drift levels might be smaller than the level detectable with the 1–3 m spatial resolution offered by the current generation of SAR sensors. With the present resolution of the DPM being around 30 m x 30 m per pixel, the proposed method is more likely to be useful for detecting large damaged buildings, damaged building aggregates, and damage to dense building blocks, more than damage to isolated smaller buildings. With the advent of commercial SAR satellite constellations like Capella Space and ICEYE, sub-meter SAR imagery is becoming available. Digital elevation maps (DEMs), which are used in terrain correction or removal of topographically induced phase variations in the generation of the DPMs, are also now becoming available at meter and sub-meter resolutions. Thus, some of the aforementioned deficiencies should be addressable.

# 4 Conclusions

This article describes a framework for semi-automated damage assessment due to earthquake from Earth Observation (EO) data and other supplementary datasets, leveraging upon recent advances in machine-learning algorithms. This framework combines high-resolution building inventory data from OpenStreetMap and other local sources with image-processing algorithms for the detection of earthquake damage using InSAR data generated by the JPL-ARIA initiative, along with supplementary geospatial datasets as inputs to a random forest ML classification model. The ML model is trained using detailed building damage datasets from past events in a supervised learning framework. Both multi-class and binary damage classification are attempted and we compared the predicted damage labels with ground truth damage grade labels reported in field surveys. Binary damage classification is shown to outperform multi-class classification for the earthquakes studied, and the highest classification accuracy scores are observed for the case where the largest number of building attributes relevant for structural damage are available.

The time-span between the acquisition of the pre-event and post-event images can have a considerable impact on the potential false positives. The closer the "before" and "after" bracket the event, the fewer the false positives that are likely to be observed. Multiple SAR satellite missions currently have revisit intervals of a few days. DPMs are already reliably generated by the ARIA team within a few days of major earthquakes. With the planned launches of the NASA-ISRO SAR (NISAR) mission (Kellogg et al., 2020) and ALOS-4 (Motohka et al., 2020), and the advent of commercial SAR satellite constellations Capella Space (2022); ICEYE (2022), post-event image acquisition are expected to become available within 1-2 days after observation or even within a few hours in response to disasters (Kellogg et al., 2020). Earthquake ground shaking intensity maps are also made available by the US Geological Survey within a few hours after an event. The curation of building inventory datasets is a critical step that should ideally be undertaken before the occurrence of a disaster event and such datasets should be regularly updated. In the absence of a precompiled building inventory dataset for the affected region, building extracts from OSM can be used within the proposed damage detection framework. The training of the machine learning models would have been undertaken prior to the disaster event, and the trained model can then be deployed for damage detection following an earthquake as soon as the pre-event building inventory, ShakeMap, and DPM become available. Thus, the time-frame for obtaining the first results from the proposed damage detection framework is expected to be in the order of 1–7 days following an earthquake.

Machine learning models for the prediction of post-disaster damage can benefit greatly from having access to labelled and georeferenced building damage data. Cross-regional training datasets will also help greatly improve the performance of these models for earthquakes in new regions previously unseen by the model. By expanding the datasets used to train the ML damage classification models, we can transfer the learning from regions with more damage data availability to data sparse regions. Cross-regional training is also critical as it will ultimately make such damage classification models more robust as they can be more confidently applied to future disasters, which may affect regions the model has not been trained on. This remains a challenge, however, particularly in the case of multi-class classification of damage grades, due to differences in the damage scales used in the field damage surveys in different events, and also because of the subjectivity involved in the assignment of damage grades by the field damage surveyors.

One of the eventual promises of the framework described in this paper is to be able to predict damage using InSAR data even for locations that aren't present in the training data. Ideally, region-specific damage detection models could be developed that take into consideration input features that are region-specific. Alternatively, region-specific or location-specific characteristics could be encoded as additional input features to a global remote-sensing based damage detection model. For instance, one of the inputs in the proposed methodology is the ground shaking intensity map (ShakeMap) generated by the US Geological Survey, which does take into consideration local site conditions, albeit through the proxy measure of Vs30 values. The tectonic setting is also taken into account implicitly in the derivation of the ShakeMap, as the choice of the ground motion model used to predict the ground shaking intensities in the affected area depends on the tectonic region type. If information about building construction types is available, this can be encoded as a categorical input feature, as was done for the 2015 Gorkha and 2017 Puebla examples in this study.

Both SAR and optical EO data have their relative strengths. SAR data can be obtained even in poor-light conditions, at night, and independent of cloud cover. However, while meter-level spatial resolution is now offered by several SAR satellites, optical EO data is currently available at sub-meter spatial resolution from several satellites, making it particularly appealing for building-level damage detection methods. Finer differentiation of damage grades involving detection of cracks in walls or residual drifts is still quite challenging with the 1–3 m spatial resolution offered by the current generation of SAR sensors. Thus, simultaneous use of SAR and optical EO data in a deep learning workflow could potentially combine the advantages of the two different data types and increase the accuracy of damage detection. The addition of supplementary information, such as hazard intensity data measured at a few locations, local site conditions, and building attribute data could also help improve rapid post-earthquake damage classification.

*Code availability.* The Python code and Jupyter notebooks used for the analysis are available at https://github.com/gemsciencetools/eo-damage-detection under the GNU Affero General Public License (v3.0).

*Data availability.* The data sources used for the case studies are listed in Table 1.

*Author contributions.* AR performed the damage analyses and drafted the manuscript with contributions from all authors. JJ processed the SAR data and produced the DPMs. VS conceptualized the project goals, supervised the research activity planning and execution, and provided critical review of the manuscript drafts. GM led the technical oversight of the project, and also provided critical review of the manuscript drafts. SY provided guidance on the overall design of the project regarding event identification criteria and production/usage of the DPMs.

*Competing interests.* The authors declare that they have no conflict of interest.

*Acknowledgements.* The authors would like to thank Suman Pradhan (National Society for Earthquake Technology – Nepal) for kindly sharing the detailed building-level damage data for Budhanilkantha municipality of Kathmandu for the 2015 Gorkha earthquake. This paper is the outcome of a pilot study on earthquake and flood damage detection using remote sensing data that was financially supported by the World Bank, which also provided technical supervision of the project. We would also like to thank Matthew Foote for several useful discussions during the early stages of this work. Part of the research was carried out at the Jet Propulsion Laboratory, California Institute of Technology, under a contract with the National Aeronautics and Space Administration. Part of this research was carried out at the Earth Observatory of Singapore via its funding from the Nanyang Technological University Award #021255-00001 (EOS Contribution Number 413).

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

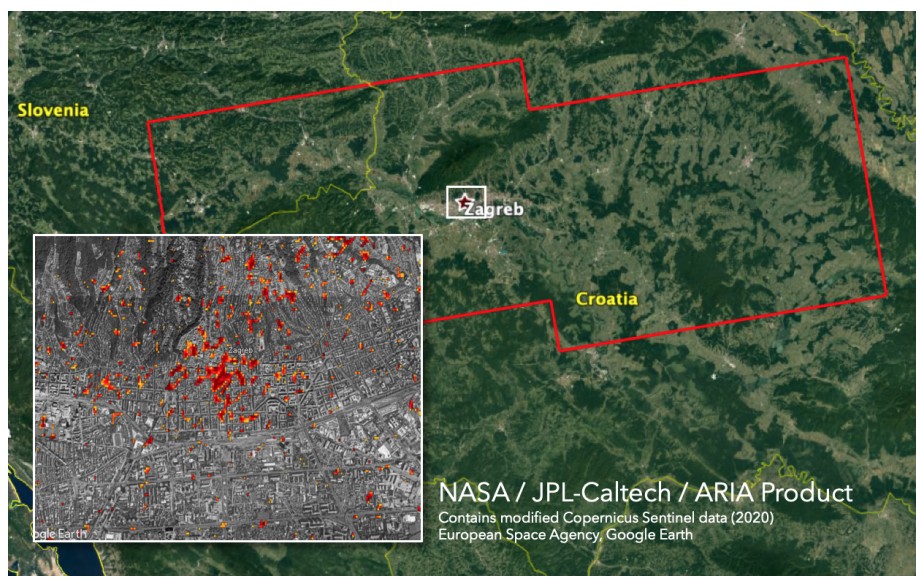

**Figure 1.** DPM for the March 2020 Zagreb earthquake (Contains modified Copernicus Sentinel data and © Google Earth 2020 imagery, processed by NASA / JPL-Caltech)

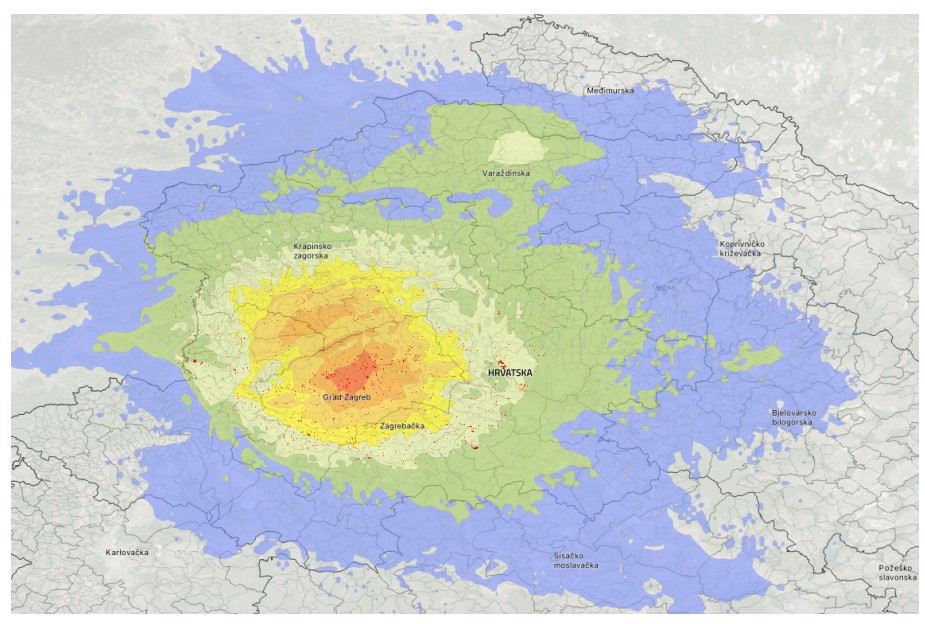

**Figure 2.** USGS ShakeMap for the March 2020 Mw5.3 Zagreb earthquake

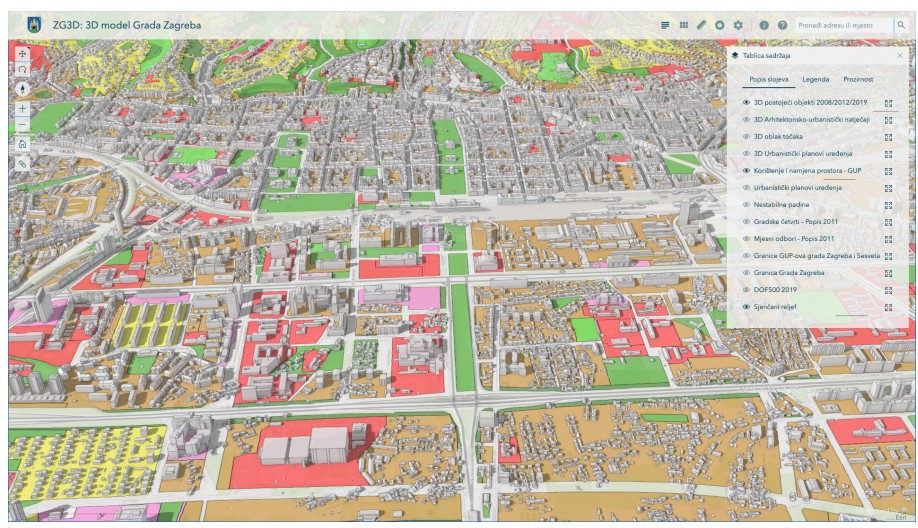

**Figure 3.** Building inventory dataset for the city of Zagreb (Source: Office for Strategic Planning and City Development, Grad Zagreb. URL: https://zagreb.gdi.net/zg3d/)

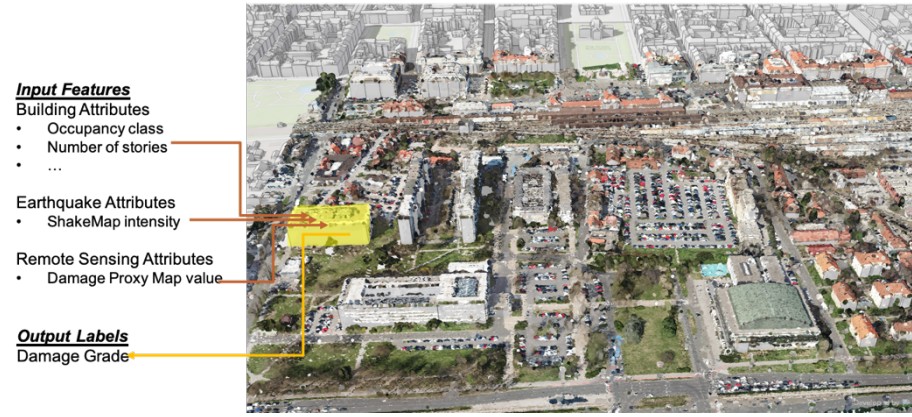

**Figure 4.** Illustration of the input feature vector and output label vector for a selected building in Zagreb (Map source: Office for Strategic Planning and City Development, Grad Zagreb. URL: https://zagreb.gdi.net/zg3d/)

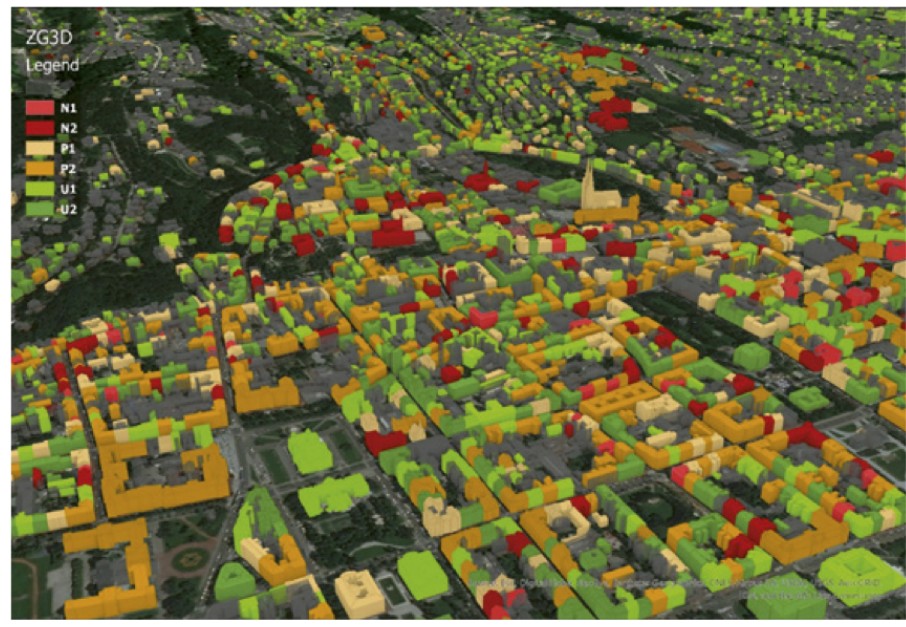

*Source: University of Zagreb Faculty of Civil Engineering.*
*Notes: 3D model of buildings represented by the usability classifications color (green, yellow, and red).*

**Figure 5.** Building damage grades in the city of Zagreb following the 2020 March earthquake

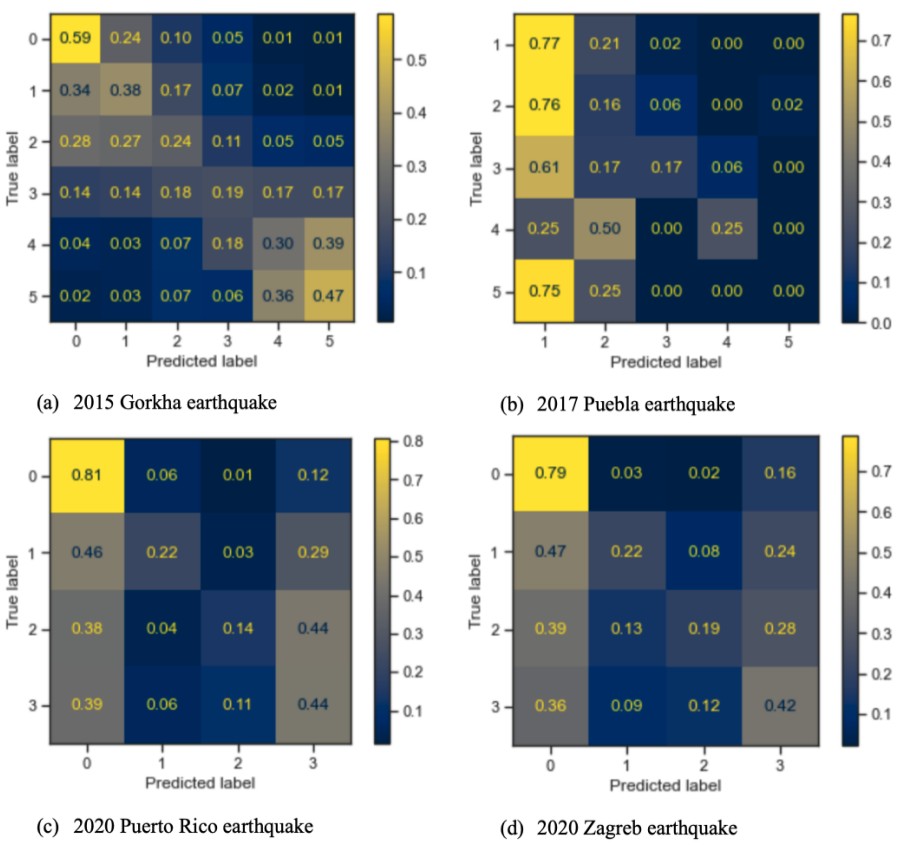

**Figure 6.** Normalized confusion matrices for the test set for multi-class damage classification. The labels refer to progressive damage grades, as described in Tables 2, 4, 6, and 8.

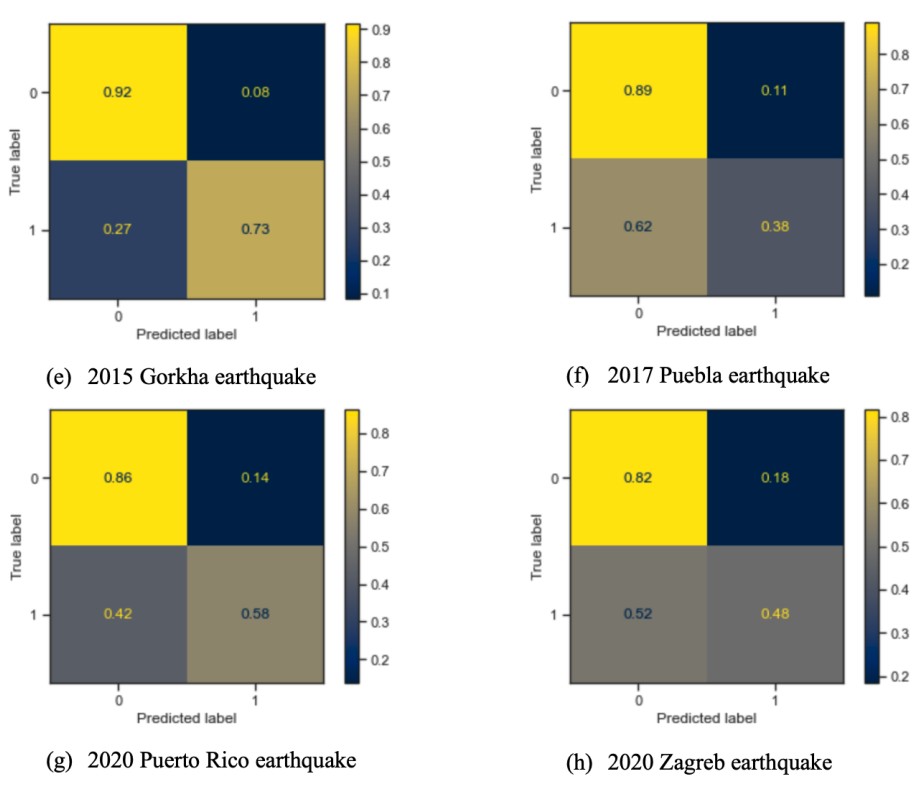

**Figure 7.** Normalized confusion matrices for the test set for binary damage classification. Label 0 refers to undamaged buildings and label 1 refers to damaged buildings.

**Table 1.** Data sources and available input features for the case study earthquakes

| Input feature sources and attributes | 2015 Gorkha earthquake | 2017 Puebla earthquake | 2020 Puerto Rico earthquake | 2020 Zagreb earthquake |
|---|---|---|---|---|
| **Date, magnitude of event, and country** | 25 April 2015, M7.8, Nepal | 19 September, 2017, M7.1, Mexico | 7 January, 2020, M6.4, United States | 22 March, 2020, M5.3, Croatia |
| **Affected buildings** | 750,000 buildings damaged, of which 500,000 completely destroyed | Over 3,000 buildings damaged, and 38 buildings collapsed | 335 buildings damaged, of which 77 completely destroyed | 26,000 homes damaged or destroyed |
| **Source of building footprints** | OSM | INEGI | OSM | Grad Zagreb |
| **Available building attributes** | Number of stories, age, construction type, primary occupancy, structural irregularities, floor type, roof type, adjacent buildings, slope of the ground | Number of stories, age, construction type, soil period | None | Number of stories, primary occupancy |
| **Source of ground shaking intensity map** | USGS ShakeMap | USGS ShakeMap | USGS ShakeMap | USGS ShakeMap |
| **Source of remote sensing damage proxy** | ARIA DPM | ARIA DPM | ARIA DPM | ARIA DPM |
| **Source of ground truth damage labels** | NSET (subset covering Budhanilkantha municipality of Kathmandu) | Buendía Sánchez and Angulo (2017) & Reinoso et al. (2021) | FEMA | University of Zagreb, Faculty of Civil Engineering |
| **Number of damage grades (*excluding "No damage"*)** | Five (EMS-98) (Slight, moderate, substantial, very heavy, destruction) | Five (Slight, intermediate, heavy, partial collapse, total collapse) | Three (Slight, moderate, heavy) | Three (Green, yellow, red) or (Slight, moderate, heavy) |
| **Damage grades considered as "*Damaged*" for binary classification** | Substantial damage, very heavy damage, destruction | Heavy damage, partial collapse, total collapse | Moderate damage, heavy damage | Moderate damage, heavy damage |

**Table 2.** Test-set performance metrics for multi-class classification for the 2015 Gorkha earthquake

| Damage Grade | Precision | Recall | F1-score | Support |
|---|---|---|---|---|
| 0. No Damage | 0.62 | 0.59 | 0.60 | 2,046 buildings |
| 1. Slight Damage | 0.46 | 0.38 | 0.42 | 1,567 buildings |
| 2. Moderate Damage | 0.20 | 0.24 | 0.22 | 557 buildings |
| 3. Heavy Damage | 0.14 | 0.19 | 0.16 | 263 buildings |
| 4. Very Heavy Damage | 0.21 | 0.30 | 0.24 | 171 buildings |
| 5. Destruction | 0.34 | 0.47 | 0.39 | 183 buildings |
| Accuracy | | | 0.44 | 4,787 buildings |
| Macro average | 0.33 | 0.36 | 0.34 | 4,787 buildings |
| Weighted average | 0.47 | 0.44 | 0.45 | 4,787 buildings |
| Balanced accuracy | | | 0.36 | 4,787 buildings |

**Table 3.** Test-set performance metrics for binary classification for the 2015 Gorkha earthquake

| Damage Grade | Precision | Recall | F1-score | Support |
|---|---|---|---|---|
| 0. Undamaged | 0.96 | 0.92 | 0.94 | 4,170 buildings |
| 1. Damaged | 0.57 | 0.73 | 0.64 | 617 buildings |
| Accuracy | | | 0.89 | 4,787 buildings |
| Macro average | 0.76 | 0.82 | 0.79 | 4,787 buildings |
| Weighted average | 0.91 | 0.89 | 0.90 | 4,787 buildings |
| Balanced accuracy | | | 0.82 | 4,787 buildings |

**Table 4.** Test-set performance metrics for multi-class classification for the 2017 Puebla earthquake

| Damage Grade | Precision | Recall | F1-score | Support |
|---|---|---|---|---|
| 1. Slight Damage | 0.66 | 0.77 | 0.71 | 142 buildings |
| 2. Intermediate Damage | 0.24 | 0.18 | 0.20 | 51 buildings |
| 3. Heavy Damage | 0.36 | 0.22 | 0.28 | 18 buildings |
| 4. Partial Collapse | 0.00 | 0.00 | 0.00 | 4 buildings |
| 5. Total Collapse | 0.00 | 0.00 | 0.00 | 4 buildings |
| Accuracy | | | 0.56 | 219 buildings |
| Macro average | 0.25 | 0.23 | 0.24 | 219 buildings |
| Weighted average | 0.51 | 0.56 | 0.53 | 219 buildings |
| Balanced accuracy | | | 0.23 | 219 buildings |

**Table 5.** Test-set performance metrics for binary classification for the 2017 Puebla earthquake

| Damage Grade | Precision | Recall | F1-score | Support |
|---|---|---|---|---|
| 0. Undamaged | 0.92 | 0.91 | 0.91 | 193 buildings |
| 1. Damaged | 0.36 | 0.38 | 0.37 | 26 buildings |
| Accuracy | | | 0.84 | 219 buildings |
| Macro average | 0.64 | 0.65 | 0.64 | 219 buildings |
| Weighted average | 0.85 | 0.84 | 0.85 | 219 buildings |
| Balanced accuracy | | | 0.65 | 219 buildings |

**Table 6.** Test-set performance metrics for multi-class classification for the 2020 Puerto Rico earthquake

| Damage Grade | Precision | Recall | F1-score | Support |
|---|---|---|---|---|
| 0. No Damage | 1.00 | 0.81 | 0.89 | 185,139 buildings |
| 1. Slight Damage | 0.01 | 0.22 | 0.01 | 329 buildings |
| 2. Moderate Damage | 0.01 | 0.14 | 0.01 | 107 buildings |
| 3. Heavy Damage | 0.00 | 0.44 | 0 | 18 buildings |
| Accuracy | | | 0.81 | 185,593 buildings |
| Macro average | 0.25 | 0.40 | 0.23 | 185,593 buildings |
| Weighted average | 1.00 | 0.81 | 0.89 | 185,593 buildings |
| Balanced accuracy | | | 0.40 | 185,593 buildings |

**Table 7.** Test-set performance metrics for binary classification for the 2020 Puerto Rico earthquake

| Damage Grade | Precision | Recall | F1-score | Support |
|---|---|---|---|---|
| 0. Undamaged | 1.00 | 0.86 | 0.93 | 185,468 buildings |
| 1. Damaged | 0.00 | 0.58 | 0.01 | 125 buildings |
| Accuracy | | | 0.86 | 185,593 buildings |
| Macro average | 0.50 | 0.72 | 0.47 | 185,593 buildings |
| Weighted average | 1.00 | 0.86 | 0.93 | 185,593 buildings |
| Balanced accuracy | | | 0.72 | 185,593 buildings |

**Table 8.** Test-set performance metrics for multi-class classification for the 2020 Zagreb earthquake

| Damage Grade | Precision | Recall | F1-score | Support |
|---|---|---|---|---|
| 0. No Damage | 0.94 | 0.79 | 0.86 | 75,118 buildings |
| 1. Slight Damage | 0.36 | 0.22 | 0.27 | 6,431 buildings |
| 2. Moderate Damage | 0.13 | 0.19 | 0.16 | 1,694 buildings |
| 3. Heavy Damage | 0.01 | 0.42 | 0.02 | 432 buildings |
| Accuracy | | | 0.73 | 83,675 buildings |
| Macro average | 0.36 | 0.41 | 0.33 | 83,675 buildings |
| Weighted average | 0.87 | 0.73 | 0.79 | 83,675 buildings |
| Balanced accuracy | | | 0.40 | 83,675 buildings |

**Table 9.** Test-set performance metrics for binary classification for the 2020 Zagreb earthquake

| Damage Grade | Precision | Recall | F1-score | Support |
|---|---|---|---|---|
| 0. Undamaged | 0.98 | 0.82 | 0.89 | 81,549 buildings |
| 1. Damaged | 0.06 | 0.48 | 0.11 | 2,126 buildings |
| Accuracy | | | 0.81 | 83,675 buildings |
| Macro average | 0.52 | 0.65 | 0.50 | 83,675 buildings |
| Weighted average | 0.96 | 0.81 | 0.87 | 83,675 buildings |
| Balanced accuracy | | | 0.65 | 83,675 buildings |

**Table 10.** Summary of balanced accuracy scores for multi-class and binary classification

| Balanced accuracy scores | 2015 Gorkha earthquake | 2017 Puebla earthquake | 2020 Puerto Rico earthquake | 2020 Zagreb earthquake |
|---|---|---|---|---|
| **Binary classification** | 0.82 | 0.65 | 0.72 | 0.65 |
| **Multi-class classification** | 0.36 | 0.23 | 0.40 | 0.40 |