# Peer review of "Earthquake Building Damage Detection based on Synthetic Aperture Radar Imagery and Machine Learning"

_Natural Hazards and Earth System Sciences, 2022_

## Author Response (AR1)

Overall, a very well written paper that will appeal to earthquake engineers active in building damage assessment and the wider audience interested in the use of novel technologies (InSAR) and machine learning. Nevertheless, some points would benefit from additional explanations/clarifications. My comments are provided below.

We would like to thank the reviewer for highly constructive comments on the article. The suggestions are highly appreciated. Our responses to the comments are provided below.

Comment 1

Lines 175-176: "The SAR-derived DPMs published by the ARIA project are used as the primary remote-sensing proxy to identify surface-level changes that are potentially attributable to earthquake-induced building damage."

Could you please clarify how you interpret the correlation between surface-level changes and earthquake-induced building damage? Would it be possible to have seismic building damage without significant change in the ground surface level? If so, how would you proceed to remotely detect building damage using InSAR or EO?

==

In previous events where earthquake DPMs were generated by the ARIA team, such as the 2015 Gorkha earthquake in Nepal and the 2016 Central Italy earthquakes, the DPMs were found to have good correlation with the actual damage mapped in field surveys [Yun et al. (2015), Sextos et al. (2018)]. However, landslides and rockfall can also lead to surface-level changes, and so can changes in vegetation and other phenomena such as building construction. Thus, while attempting to detect building damage, care needs to be exercised to limit the focus of the DPMs to locations where buildings are known to exist. The time-span between the acquisition of the pre-event and post-event images can also have a considerable impact on the potential false positives. The closer the 'before' and 'after' bracket the event, the fewer the false positives that are likely to be observed.

It is also certainly possible to have seismic building damage without observing significant change in the ground surface level. Damage to internal walls or columns that may have severely compromised the structural integrity of the building without causing externally visible damage or collapse may not be detectable through remote sensing. Storey drifts of 2% for braced steel structures or concrete shear wall structures may be technically classified as 'collapsed' (eg. FEMA 356), but such drift levels might be smaller than the level detectable with the 1–3 m spatial resolution offered by the current generation of SAR sensors. Dong and Shan (2013) provide a good review of previous studies that investigated the relationship between building appearance in remote sensing data and building damage grades. In general, they concluded that the higher damage states like complete collapse are more detectable through remote sensing data, but lower damage states are more challenging to detect. With the advent of commercial SAR satellite constellations like Capella Space and ICEYE, sub-meter SAR imagery is becoming available, and some of these deficiencies should be addressable.

In recognition of these limitations in the SAR and EO datasets, in our study we have proposed to incorporate other variables (such as building attributes or the expected macroseismic intensity at the location of the building) to mitigate these issues. We have updated this section of the manuscript to clarify these limitations, and how we propose to address them.

==

Comment 2

Lines 185-188: "The problem presented is one of multi-class classification, and two ensemble machine-learning classification algorithms are employed—the Random Forest classifier for the cases involving only numeric features, and Histogram-Based Gradient Boosting classifier for the cases which also involve categorical features amongst the selected building attributes."

Could you please explain why you selected the random forest algorithm and histogram-based gradient boosting classifier? Did you try any other algorithms? How did they perform?

==

In a preliminary phase, we compared different algorithms that permit multiclass classification, including support vector machines, k-nearest neighbours, Naive Bayes, and Random Forest. Since the problem of damage classification typically involves highly imbalanced datasets, where the buildings in "no damage" state dominate the buildings in all other damage states, often by multiple orders of magnitude, all of the above classifier algorithms tended to overlearn the label with the higher number of training examples (i.e., "no damage"). The Random Forest algorithm was eventually selected for the study as it allows for the assignment of weights to the training examples. The training examples in each damage class were then weighted in inverse proportion to the class frequencies observed in the input data, in order to better handle the class imbalance in the input damage datasets. The Histogram-Based Gradient Boosting classifier was preferred in the cases where categorical features were present amongst the selected building attributes, in addition to purely numerical features. This was because the Histogram-Based Gradient Boosting classifier provides native categorical support, which helps avoid one-hot encoding to transform categorical features as numeric arrays. We have included this explanation in the revised version of the manuscript.
==

Comment 3

Lines 188-189: "The models are trained with a 70% subset of the available data, and then the best-fit models are tested against the 30% hold-out subset."

Could you please explain why you chose a 70%/30% for the training and testing set? Did you try 80%/20% for example? Was 70%/30% giving the best performance?

==

There is certainly a tradeoff between using more samples for training the algorithm versus reserving sufficient samples for the test set. Using a higher fraction of the available data for training can result in overfitting. Previous empirical studies have demonstrated that using 70-80% of the data for training and reserving 20-30% of the data for testing yields optimal results in terms of improving the accuracy of the model while minimizing the tendancy for overfitting [see Gholamy et al. (2018) for instance]. The decision to choose a 70%/30% split for the training and testing set (say, over an 80%/20% split) was ultimately driven by the paucity of 'collapse' labels in the damage datasets, particularly for the 2017 Puebla event where reserving only 20% of the dataset for testing would leave very few 'collapse' labels in the test set to evaluate the accuracy of the fitted model.
We have added this explanation in the revised manuscript, with references that also justify this approach.
==

Comment 4

Lines 211-213: "Another important reason to undertake a binary damage classification exercise is that it permits the aggregation of building damage datasets from different events into a larger training pool."

Could you please clarify what you understand under a "larger training pool" for a machine learning model?
The paucity of building damage data that can be used for training is one of the main challenges affecting machine learning models for damage prediction. Different countries use different methodologies and different damage scales to assess building damage following earthquakes. While the definition of the lower damage grades might differ considerably between different scales, collapse is often consistently defined. Thus, if the focus is restricted to identifying collapsed buildings from non-collapsed buildings, a wider set of events from the region can be used train the model, given that the training labels in this case coming from different events will be consistently defined. We have included this information in the revised version of the manuscript.

Lines 325-329: "Cross-regional training datasets will also help greatly improve the performance of these models for earthquakes in new regions previously unseen by the model. By expanding the datasets used to train the ML damage classification models, we can transfer the learning from regions with more damage data availability to data sparse regions. Cross-regional training is also critical as it will ultimately make such damage classification models more robust as they can be more confidently applied to future disasters, which may affect regions the model has not been trained on."

Could you comment on the performance of a ML model applied to region on which it has not been trained on? When creating a "larger training pool", could you please explain how you would capture regional specificities?

This reviewer is of the opinion that each location/region has specificities (e.g., construction practices, seismic setting, ground conditions) and thus has concerns regarding the applicability of a "one-fits-all" ML damage prediction model.

==

The location/region-specific concerns expressed by the reviewer are well appreciated. One of the eventual promises of the framework described in this paper is to be able to predict damage using InSAR data even for locations that aren't present in the training data. Ideally, region-specific damage detection models could be developed that take into consideration input features that are region-specific. Alternatively, region-specific or location-specific characteristics could be encoded as additional input features to a global remote-sensing based damage detection model. For instance, one of the inputs in the proposed methodology is the ground shaking intensity map (ShakeMap) generated by the US Geological Survey, which does take into consideration local site conditions, albeit through a proxy measure (Vs30). The tectonic setting is also taken into account implicitly in the derivation of the ShakeMap, as the choice of the ground motion model used to predict the ground shaking intensities in the affected area depends on the tectonic region type. If information about building construction types is available, this can be encoded as a categorical input feature, as was done for the 2015 Gorkha and 2017 Puebla examples in this study. Other researchers such as Moya et al. (2018) have also attempted to incorporate fragility functions into a machine learning

framework. The concerns raised by the reviewer are also valid for the current state-of-practice in earthquake risk assessment, where ground motion models or empirical fragility / vulnerability models that have been derived for a particular region where sufficient data are available, are routinely employed for damage / loss assessment in other regions (ideally sharing similar characteristics) where not enough data are available. We have included a discussion about this limitation and potential source of bias in the concluding remarks.
==

Comment 5

Lines 320-323: " The training of the machine learning models happens prior to the disaster event, and the trained model can be deployed for damage detection following an earthquake as soon as the pre-event building inventory, ShakeMap, and DPM become available ".

Could you please clarify the process? Why does this information only appear in the conclusion? Did you try to train a ML model for a region prior to a disaster and test the ML model after the earthquake event?

==
This statement is meant to depict how the proposed framework would work in a real-time post-event damage assessment environment. Within the scope of the current study, we unfortunately did not come across building-level damage data from multiple events within the same country or geographic region that could be used to train a ML model for the region using data from previous events prior to a disaster and test the ML model after the subsequent earthquake event. The phrasing of the sentence has been improved to better convey the intention, to: "*The training of the machine learning models would have been undertaken prior to the disaster event, and the trained model can then be deployed for damage detection following an earthquake as soon as the pre-event building inventory, ShakeMap, and DPM become available*"
==

Comment 6

Lines 342-343: "Code availability. The Python code and Jupyter notebooks used for the analysis are available at https://github.com/gemsciencetools/eodamage-detection under the GNU Affero General Public License (v3.0)."

GitHub repo accessed by the reviewer on 29 May 2022

Data and code were found for the Gorkha, Puebla, and Zagreb earthquakes, as well as the central Italy earthquakes. However, no folder/code related to the Puerto Rico earthquake could be found.

==
The folder containing the data and code related to the Puerto Rico earthquake has been added to the repository.
==
* * *
**Reviewer 2**

The paper describes a framework for building damage classifications after an earthquake that combines InSAR data, high-resolution building inventory data and earthquake ground shaking intensity maps. The classification is performed using two different strategies (multi-class and bynary classification) and two different machine learning algorithms.

I am a researcher in the field of computer science, particularly in machine learning. Therefore, this review will focus on issues related to machine learning techniques used to solve a problem in the area of natural disasters.

The article is well structured and organised. It is not difficult to understand the main ideas of the work done and it is well written. The work described represents a very valid proposal but the original contributions are minor. Anyway, if the proposed framework proposed is well evaluated and validated, for me, it can be accepted for publication. However, as the paper is, in terms of validation, does not follows the standards and requirements of this journal.

The authors claim innovation in the use of the Multi-class damage grade classification using InSAR data. I have nothing against but the results presents are weak to validated the method. I think authors should present results that clearly show the benefits of the strategy used when compared to other techniques proposed. The multi-class strategy seems to me very important but the results are weak and are not compared. Also, the selection building damage categories is a critical point, regarding the urgency of action. This should be more discussed and explained in the paper.

Deep learning techniques are been used to solve many problems in many fields and presenting good results or better than the previous methods. I found it very strange that the authors would talk about recent advances in machine learning and then not use deep learning techniques for classification.

I understand that in some situations it can be difficult to use deep learning techniques, particularly when there is not much data available for training. If that's the case, I think the authors should present results demonstrating this. Also, the data augmentation methods should be considered.

==

The comments made by the reviewer are well taken. We would like to emphasize that the focus of the article is more about the integration of multiple input datasets including InSAR data for building damage detection, rather than on machine learning itself.

As far as we are aware, multi-class damage classification at the building level using SAR data has not been attempted before, so we are unfortunately unable to compare our results with any existing literature, and this was precisely one of the reasons we pursue this study. It was also our intention to propose an open framework and data that other earthquake engineers and risk modellers could use for rapid loss assessment.

Deep learning techniques are more suited for structured data (images, audio, text) with large sample sizes, while the datasets used in this study are tabular (each row representing one building unit) and are small or medium sized (typically of the order of 1,000–10,000 samples). While it's true that for tabular data, both decision forest based approaches and deep neural networks could be used, as the reviewer acknowledges, deep learning techniques perform better with large sample sizes, which was unfortunately not the case for this study, given the limited availability of building-level damage datasets and the limited number of labelled damaged buildings within each dataset. Grinsztajn et al. (2022) conclude that for medium sized tabular data (~10,000 samples), tree-based models outperform deep learning methods, with much less computational cost. Similarly, Xu et al. (2021)

also conclude that forests perform better than deep neural nets for tabular data with small sample sizes.

The reviewer's comment about the selection of appropriate building damage categories is apt. Dell'Acqua and Gamba (2012) highlight the need to develop damage scales specific to earth observation based damage assessments, possibly tied to existing damage scales that are widely used for field surveys of building damage (such as EMS 98). Cotrufo et al. (2018) propose a building damage assessment scale tailored for optical satellite imagery and aerial imagery. However, a similar damage scale tailored for InSAR based building damage assessment is still lacking. The revised manuscript discusses these limitations and potential areas for future research in the concluding remarks.

**References cited in this response**

Cotrufo, S., Sandu, C., Giulio Tonolo, F., & Boccardo, P. (2018). Building damage assessment scale tailored to remote sensing vertical imagery. European Journal of Remote Sensing, 51(1), 991–1005. https://doi.org/10.1080/22797254.2018.1527662

Dell'Acqua, F., & Gamba, P. (2012). Remote sensing and earthquake damage assessment: Experiences, limits, and perspectives. Proceedings of the IEEE, 100(10), 2876–2890. https://doi.org/10.1109/JPROC.2012.2196404

Grinsztajn, L., Oyallon, E., & Varoquaux, G. (2022). Why do tree-based models still outperform deep learning on tabular data? https://arxiv.org/abs/2207.08815

Xu, H., Kinfu, K. A., LeVine, W., Panda, S., Dey, J., Ainsworth, M., Peng, Y.-C., Kusmanov, M., Engert, F., White, C. M., Vogelstein, J. T., & Priebe, C. E. (2021). When are Deep Networks really better than Decision Forests at small sample sizes, and how? https://arxiv.org/abs/2108.13637v4

---

## Author Response (AR2)

**Anonymous Referee #3 (Report #1)**

The proposed work focuses on the development of a framework for building-level earthquake damage assessment. High-resolution SAR data are used as input together with building-related datasets and earthquake-related intensity maps for the classification of the building damage states. The paper is well-written and the presence of a complete repository with data and codes to evaluate the proposed methods is of great value.

We would like to thank the reviewer for highly constructive comments on the article. The suggestions are highly appreciated. Our responses to the comments are provided below.

The adopted ML technique for the implementation of the classifier has been justified by looking into the literature where similar solutions have been successfully adopted and compared with other approaches. Due to the presence of significative differences between the proposed method, in particular in terms of combined input data, and the previous works, other ML techniques should be used for comparison. This aspect should be mentioned in the manuscript.

>>> In lines 182-190 of the revised manuscript, we explain why deep-learning techniques were ruled out for the current study. In lines 191-202, we also mention having compared other ML techniques in the preliminary stage of the study and the reason for settling on the Random Forest classifier and Histogram-Based Gradient Boosting classifier for the remainder of the study.

The selection of an ML technique includes the definition of multiple hyperparameters that can be modified/optimized to maximize the system's performance. This point has not been discussed at all in the work. Even if default hyperparameters have been used, they have to be presented and future optimization solutions can be considered.

>>> We have included the following text in the revised manuscript to discuss the treatment of hyperparameters, starting at line 203.

The next step involves tuning of the hyper-parameters of the chosen classifier algorithms, where hyper-parameters are the model parameters that are not directly learnt during the training phase. Probst et al. (2019) provide a thorough overview of the hyper-parameters and tuning strategies for the random forest algorithm. Random forest algorithms have three main hyper-parameters, including the number of trees in the forest, the node size, and the number of features sampled when looking for the best split for a node. The number of features sampled at each split is set to the square root of the number of predictor variables, which Probst et al. (2019) indicate as a reasonable value for low-dimensional classification problems. Optimal values for the number of trees in the forest and the node size are obtained through an exhaustive grid search strategy, to pick the combination of hyper-parameter values that result in the best cross validation score. For all other hyper-parameters which have less of an impact on the model performance, we use the default values provided by the software package scikit-learn (Pedregosa et al., 2011).

Even if the datasets are unbalanced, proper strategies have been considered to overcome this problem. However, in Fig. 6 it is evident that the learned classifier is significantly biased in the prediction of low-number labels. This aspect needs to be further discussed and potential solutions to overcome this problem should be proposed. The reported results for a multi-class classifier are barely useful, is it a problem of the method or mainly due to the available data? The possibility to compare the obtained results with other baseline methods could have answered in part to this question.

>>> The observation that the learned classifier tends to be biased in the prediction of lower damage grades is true, and this is particularly evident in the case of the Puebla earthquake results in Figure 6b. As mentioned in the manuscript, there aren't many previous studies that have attempted multi-class damage classification for earthquake building damage that we can compare our results with.

Mangalathu et al. (2020) report that in their attempt to classify building damage from the 2014 South Napa earthquake into three damage classes, the random forest algorithm was correctly able to identify only 12.5% of the red-tagged buildings. Using the 2016 Kumamoto earthquake as a case study for binary damage classification, Bai et al. (2017) obtained a prediction accuracy of 38.9% for identifying damaged buildings when multi-temporal post-event SAR images were used along with the K-Nearest Neighbours learning algorithm. Lanteri et al. (2017) report that for the 24 August 2017 Central Italy earthquake, the Copernicus Emergency Management Service's damage grading maps for the event, made by comparing pre- and post-event optical satellite images, correctly predicted 18.85% of the highly damaged or completely destroyed buildings in the affected areas. In comparison to these previous studies, the prediction accuracy for the highest damage grade in the current study ranges from 42% to 47% for the 2020 Zagreb, 2020 Puerto Rico, and 2015 Gorkha earthquakes. We contend that these results are still quite useful, as they are able to correctly identify over 40% of the heavily damaged or collapsed buildings. We have also included the following text in the revised manuscript, beginning at line 355:

From Figure 6, we also observe that the true-positive prediction rates for the intermediate damage grades are lower than those for the no-damage and highest damage grades for the 2015 Gorkha earthquake and the 2020 Puerto Rico and Zagreb earthquakes. We believe that this partly stems from the fact that the existing damage scales that are widely used for field surveys of building damage, such as EMS-98 do not map directly to information available through earth observation data, particularly for the lower damage grades. For instance, the first three damage grades for reinforced concrete structures according to EMS-98 involve increasing levels of cracking in the beams and columns or partition and infill walls, and buckling of the reinforcement rods. Unless this kind of damage results in debris caused by excessive spalling of the concrete cover or partial collapse of infill walls that is visible outside the structure, these damage levels as defined in EMS-98 may be challenging to identify from EO data alone. Dell'Acqua and Gamba (2012) and Cotrufo et al. (2018) both propose a building damage assessment scale tailored for optical satellite imagery and aerial imagery. However, a similar damage scale tailored for InSAR based building damage assessment is still lacking, and merits further research.

The discussion of the obtained results is limited to the list of the classification performance obtained, further discussions should be reported. For instance, for the Gorkha earthquake, the presence of multiple building attributes was considered the main cause for the higher performance obtained. This claim can be verified by training a new model excluding this information and evaluating the performance drop. In general, the application of a sensitivity analysis could be mentioned as a useful method to better understand the role of the different input features in the evaluation of the output class.

>>> We have taken this suggestion into consideration and trained a new model for the 2015 Gorkha earthquake where the building attributes were excluded from the input feature vector. We have included the following text in the discussion section, starting from line 333

We observe that for the 2015 Gorkha earthquake, for which multiple building attributes are available for both damaged and undamaged buildings, the prediction accuracies for both binary and multi-class classification are significantly higher when compared to the earthquakes where fewer or no building attributes are available for use as input features. In order to understand the impact of including the building attributes on the performance of the classifier, we also trained the ML model for this earthquake without using any of the building attributes and limiting the input feature vector to the MMI and DPM values alone. The precision and recall for all damage grades are lower for this reduced model compared to the results reported for the full model in Table 2 and Table 3, for both multi-class classification and binary classification respectively. The recall score for the "Destruction" damage grade drops from 0.47 for the full model to 0.31 for the reduced model in the multi-class classification task, and from 0.73 to 0.45 in the binary classification task. Similarly, the balanced

accuracy score drops from 0.36 to 0.20 in the multi-class classification task, and from 0.82 to 0.59 in the binary classification task. These results clearly demonstrate the importance of including the additional building attributes in the analysis. A partial dependence analysis of the damage grade on the non-location building attribute variables for this event indicates that the building age has an impact on the damage grade, with older buildings being related to higher damage.

**Anonymous Referee #4 (Report #2)**

The manuscript is focused on the use of ML classification where different inputs are used such as shaking maps, the SAR derived DPMs map and building information with the aim of semi-automated building damage assessment due to earthquakes.
General comments:
1.The manuscript is highly focused on the description of DPMs which is a product not coming from the revised work. This section could be summarized into the data and materials.
In addition, it could be preferable to remove the section background and to summarize several of the info provided in the section 2.2 Machine learning in building damage assessment into the introduction section.
2.The data and materials could be presented in a more separated way in my opinion.
>>> We appreciate these helpful comments concerning potential improvements to the structure of the manuscript. Accordingly, we have condensed the portion of the text describing the DPM product, and now it appears under the 'Input data' subsection. The 'Background' section has also been removed, and the review of previous literature on the use of machine learning along with earth observation data for building damage assessment has been subsumed into the introduction section. The 'Input data', 'Data processing', and 'Study areas' are now presented in separate subsections in the revised manuscript.

3.In addition, the authors could present and thus discuss the problem of the different spatial resolution of the inputs…indeed, the spatial resolution of the DPMs map (which if I correctly understood is 30m) could be feasible for some buildings only… the spatial resolution of the shaking map is not discussed…
The authors could highlight that this method could be applied only to part of the urban areas where the input spatial resolution fit the average dimension of the buildings….the spatial resolution could represent a limit of this methodology
>>> We have now included the spatial resolution of the ShakeMap product (Line 146 of the revised manuscript), this is typically available on a 1km grid spacing. Each pixel of the DPM measures approximately 30m across. With the present resolution, the proposed method is indeed more likely to be useful for detecting large damaged buildings, damaged building aggregates, and damaged dense building blocks, more than damage to isolated smaller buildings. We have included a short note regarding this point in the discussions section at line 374.

4. Finally, some additional specific comments are following reported.
Please notice that line numbers refer to the manuscript "nhess-2022-125-manuscript-version3.pdf" I suppose is the clean version of the manuscript after round 1 revision
Line 142
While this study focused on implementing and testing this framework for earthquake related damage, the proposed framework adopts a modular approach…
Probably this sentence could be…
While the above mentioned studies focused on implementing and testing this framework for earthquake related damage, the proposed framework adopts a modular approach…

>>> The suggested change has been made in the revised manuscript, and now appears at line 114.

Line 190
The authors refer 2 "Thus, we clip out parts of the DPM that are outside of built-up areas, based on building footprint maps and land-use maps."
How the authors managed the high inaccuracies of SAR derived map geometry?
Also, since SAR products come from a lateral view it is well know that even if a terrain correction method is applied its radiometry content could be highly compromised (i.e. the real and imaginary parts), leading to an incorrectness of the coherence data which is the main info used to produce the DPMs maps (this effect could be more relevant for cities built in mountain areas).
Is in the DPMs metadata an info about the data accuracy (indeed, coherence reliability and spatial accuracy)? The authors could check for it and if present discuss this within the manuscript.
>>> The SAR single-look complex (SLC) images are processed using the InSAR Scientific Computing Environment (ISCE) processor developed by NASA-JPL (see Yun et al., 2015; Tay et al., 2020). The ISCE toolset (Rosen et al., 2012; Rosen et al., 2018) makes use of available information about the precise orbits of the satellite(s), atmospheric delay models, and a digital elevation model (DEM) to yield a well-aligned geocoded image. It also handles terrain correction or removal of topographically induced phase variations. The DPMs produced by Yun et al. (2015) for the 2015 Gorkha earthquake showed good correlation even in mountain areas with independent damage analyses by the National Geospatial-Intelligence Agency and the United Nations Institute for Training and Research's United Nations Operational Satellite Applications Programme. Nevertheless, coherence difference (COD) alone is indeed less effective in places with vegetation growth, agricultural activities, snowfall or rainfall, where the COD may not be due to earthquake induced damage. The DPM metadata does not typically include quantitative information about its coherence reliability and spatial accuracy. However, the readme file included with each DPM does mention that the each pixel of the DPM measures about 30 meters across and that the DPM may be less reliable over vegetated areas.

Figures 6
maybe I am in wrong but i am not sure the label's numbers (i.e 1,2,3...) were explained in the text....i suppose these refer to a damage level...
in addition, it is not clear to me way the authors are using color scales for the confusion matrices...if in the main diagonal higher values refer to higher accuracies, all other cases are omission or commission errors...
finally, the confusion matrices are not in deep discussed....supposing that the authors used a progressive labeling with the damage level (i.e. 0= no damage and 5 (or 3) = max damage) probably there are some reasons why the puebla case study shows the highest false positives... maybe it is related to the average dimension of the buildings if compared to the input's spatial resolution?
>>> The label descriptions have been added to the figure captions. The labels indeed refer to progressive damage grades. For the Puebla event, we believe the lower prediction accuracy stems primarily from the limited size of the damage dataset available for training, compared to the other three events. The average dimension of the buildings doesn't seem to be the driving factor, as we are focusing on the damages sustained in the highly urban CDMX district where the average building size is of comparable dimension to the DPM pixel size. We have included the following paragraph in the discussions section:
While four key building attributes were also available for the damaged buildings for the 2017 Puebla earthquake, the non-availability of the same for the undamaged buildings meant that a complete dataset with building attributes could not be used for the training of the ML model. Of the four events considered in this study, the 2017 Puebla event had the smallest building damage dataset available for training the ML model. Only 219 buildings in Mexico city had a complete set of building attributes and damage labels and were also covered by the DPM and ShakeMap layers, as compared to thousands or hundreds of thousands of buildings for the other three events. While the Random

Forest classification model performs well for this event in the training phase, the trained model fails to correctly identify even a single partially collapsed or totally collapsed building in the test set. Further attempts at reducing the potential overfitting of the model to the limited training data subset by adjustments to the model hyperparameters did not lead to any noticeable improvements in prediction accuracy for the event.

**References cited in this response**

Lanteri, L., Pispico, R., & Cremonini, R. (2017). Two case histories of EMS data application: Earthquake in Central Italy and flooding in Piedmont region. Copernicus EMS Mapping User Workshop 2017. Ispra, Italy.

Pedregosa, F., Varoquaux, G., Gramfort, A., Michel, V., Thirion, B., Grisel, O., Blondel, M., Prettenhofer, P., Weiss, R., Dubourg, V., Vanderplas, J., Passos, A., Cournapeau, D., Brucher, M., Perrot, M., & Duchesnay, É. (2011). Machine Learning in Python. *The Journal of Machine Learning Research*, *12*, 2825–2830. https://doi.org/10.4018/978-1-5225-9902-9.ch008

Probst, P., Wright, M. N., & Boulesteix, A. L. (2019). Hyperparameters and tuning strategies for random forest. *Wiley Interdisciplinary Reviews: Data Mining and Knowledge Discovery*, *9*(3), 1–15. https://doi.org/10.1002/widm.1301

Roseu, P. A., Gurrola, E., Sacco, G. F., & Zebker, H. (2012). The InSAR scientific computing environment. 9th European Conference on Synthetic Aperture Radar, 730–733.

Rosen, P. A., Gurrola, E. M., Agram, P., Cohen, J., Lavalle, M., & Riel, B. V. (2018). The InSAR Scientific Computing Environment 3.0: A Flexible Framework for NISAR Operational and User-Led Science Processing. 2018 IEEE International Geoscience and Remote Sensing Symposium, 4901–4904.

Tay, C. W. J., Yun, S.-H., Chin, S. T., Bhardwaj, A., Jung, J., & Hill, E. M. (2020). Rapid flood and damage mapping using synthetic aperture radar in response to Typhoon Hagibis, Japan. Scientific Data, 7(1), 1–9. https://doi.org/10.1038/s41597-020-0443-5